# Comprehensive Evaluation of Water Resources Carrying Capacity in the Han River Basin

Lele Deng [1], Jiabo Yin [1,2], Jing Tian [1], Qianxun Li [1] and Shenglian Guo [1,*]

1   State Key Laboratory of Water Resources and Hydropower Engineering Science, Wuhan University, Wuhan 430072, China; leledeng@whu.edu.cn (L.D.); jboyn@whu.edu.cn (J.Y.); jingtian@whu.edu.cn (J.T.); qianxunli@whu.edu.cn (Q.L.)
2   Hubei Provincial Key Lab of Water System Science for Sponge City Construction, Wuhan University, Wuhan 430074, China
*   Correspondence: slguo@whu.edu.cn

**Abstract:** As one of the most crucial indices of sustainable development and water security, water resources carrying capacity (WRCC) has been a pivotal and hot-button issue in water resources planning and management. Quantifying WRCC can provide useful references on optimizing water resources allocation and guiding sustainable development. In this study, the WRCCs in both current and future periods were systematically quantified using set pair analysis (SPA), which was formulated to represent carrying grade and explore carrying mechanism. The Soil and Water Assessment Tool (SWAT) model, along with water resources development and utilization model, was employed to project future water resources scenarios. The proposed framework was tested on a case study of China's Han River basin. A comprehensive evaluation index system across water resources, social economy, and ecological environment was established to assess the WRCC. During the current period, the WRCC first decreased and then increased, and the water resources subsystem performed best, while the eco-environment subsystem achieved inferior WRCC. The SWAT model projected that the amount of the total water resources will reach about 56.9 billion $m^3$ in 2035s, and the water resources development and utilization model projected a rise of water consumption. The declining WRCC implies that the water resources are unable to support or satisfy the demand of ecological and socioeconomic development in 2035s. The study furnishes abundant and valuable information for guiding water resources planning, and the core idea of this model can be extended for the assessment, prediction, and regulation of other systems.

**Keywords:** water resources carrying capacity; SWAT model; set pair analysis; sustainable development; Han River basin

## 1. Introduction

Water resources are one of the key natural resources, which are vital for feeding human beings and maintaining socioeconomic development [1]. Thanks to the rapid expansion of urbanization and industrialization, distinctive issues such as water scarcity [2] and water contamination [3] have thrown the progress of sustainable development into question in recent decades.

The concept of carrying capacity was initially proposed in the ecological community [4] and was adopted to measure the maximum of individuals maintaining a certain species in a certain area under certain conditions [5]. Nowadays, this concept has been extended into hydrological sciences and is widely used to represent the capacity of the environment or ecosystem to sustain development and specific activities.

To date, relevant research studies mainly emphasize the concept or theory establishment, and a mass of researchers have evaluated the water resources carrying capacity (WRCC) using a wide range of methods. Clarke [6] defined the WRCC as a concept that is closely related to population growth and resources consumption. Feng and Huang [7]

reckoned that the WRCC is important to measure the capacity of social development under well-living conditions. Song et al. [8] defined it as the maximum capacity for human activity in a desirable ecological system with certain socioeconomic and living levels. A host of studies also reported a variety of definitions of the concept, and a unifying concept has not yet been established [9]. From the perspective of the theory, the majority of research studies categorized WRCC as theory of sustainable development [10], and a number of new theories were introduced and extended into hydrological fields.

Several methods have been incorporated to assess carrying capacity. For instance, the ecological footprint theory has been used to evaluate the carrying capacity of resources and the sustainability of industries [11]. Nevertheless, it is arduous to employ the ecological footprint to calculate the WRCC due to variations in fluidity for water resources [12]. Chen et al. [13] developed a model based on catastrophe theory to assess the sustainable utilization of water resources, and the model might highly depend on the subjectivity of decision-makers. Focusing on the water cycle's natural and social elements, Zhang et al. [14] constructed an evaluation system that covers the dimensions of water quantity, water quality, water ecology, and water consumption. The tendency method [15], multicriteria analysis method [16], and artificial neural networks model [17] have also been utilized in relevant research studies. Meanwhile, different approaches have also been employed. Zhang et al. [18] used the principal component analysis (PCA) to analyze the river carrying capacity of the Chang-Zhu-Tan region in China and provided long-run proposals for its development. Zhao et al. [19] applied the system dynamic model to investigate the WRCC in Kunming City under four different scenarios and found that it would be extremely enhanced in 2020 under the comprehensive strategy scenario. The analytic hierarchy process was employed by Lu et al. [20] to assess the hydrological cycle in Huai'an City to achieve a coordinating balance between water resources and society. Fu et al. [21] evaluated the agricultural WRCC using the projection pursuit model in the Sanjiang Plain, China.

Although quantities of studies on the WRCC have been performed, they show deficiency in some aspects. Most current studies focused on investigating the WRCC in historical conditions. With global warming and anthropogenic changes, precipitation and runoff may be spatiotemporally redistributed, thus challenging current water resources management policies [22–24]. However, existing studies paid less attention to investigating the WRCC under future conditions. Furthermore, the methods used to evaluate the carrying capacity did not fully consider the dynamics of the WRCC system, which is highly impacted by socioeconomic development and changing environments [25].

To conquer the abovementioned deficiencies, a new methodology integrating a hydrological model, climate model, socioeconomic development model, and dynamic evaluation system is developed to investigate the WRCC in the Han River basin, China. This coupled system evaluates and projects the WRCC within a dynamic and integral framework, covering the water resources subsystem, socioeconomic subsystem, and eco-environment subsystem. This study mainly contains the following four parts: (1) establishing an evaluation index system covering three criterion layers across water resources, social economy, and ecological environment systems; (2) projecting future water resources situation in the Han River basin under climate change using the Soil and Water Assessment Tool (SWAT) hydrological model; (3) predicting water utilization situation and constructing water demand and consumption projection in the planning year; and (4) evaluating the WRCC in the Han River basin under current and future changing environments.

## 2. Study Area and Data

### 2.1. Study Area

As the largest tributary of the Yangtze River, the Han River rises in the southern Qinling Mountain. The mainstream runs through Shaanxi and Hubei provinces and flows into the Yangtze River, covering a distance of 1577 km and a catchment area of approximately 159,000 km$^2$. The basin has a great many cardinal water systems and tributaries, involving the six administrative regions of Henan, Hubei, Chongqing, Sichuan,

Shaanxi, and Gansu provinces as shown in Figure 1. The Han River basin is situated in the East Asian subtropical monsoon zone with a seasonal climate, influenced by the Eurasian cold and high pressure in the winter and the Western Pacific subtropical pressure in the summer [26]. The average annual precipitation in the Han River basin is 904 mm, which is lower than that in the Yangtze River basin. Recharged by varying precipitation, the runoff during the year shows symptoms of uneven distribution. The precipitation mainly concentrates from May to October, accounting for 55%~65% of the annual amount.

The Han River basin, a national strategic water resources security zone and pilot green development zone, has been confronted with immense problems compounding the development in this region. The impacts of climate change have exposed the basin to more varying water resources management problems, such as water supply [26] and flood control [27]. Algal blooms have taken place increasingly frequently with longer duration since 1992 [28]. Heavy metal contamination emerges downstream of Ankang City mainly owing to anthropogenic inputs, which exerts a profound influence on injured aquatic conditions [29]. Thus, policy-makers must be conversant with the WRCC for pursuing high-quality development. Carrying out research on the WRCC and making a good project for constructing the ecological–economic belt of the Han River basin are conducive to the advancement of sustainable and high-quality economic and social development.

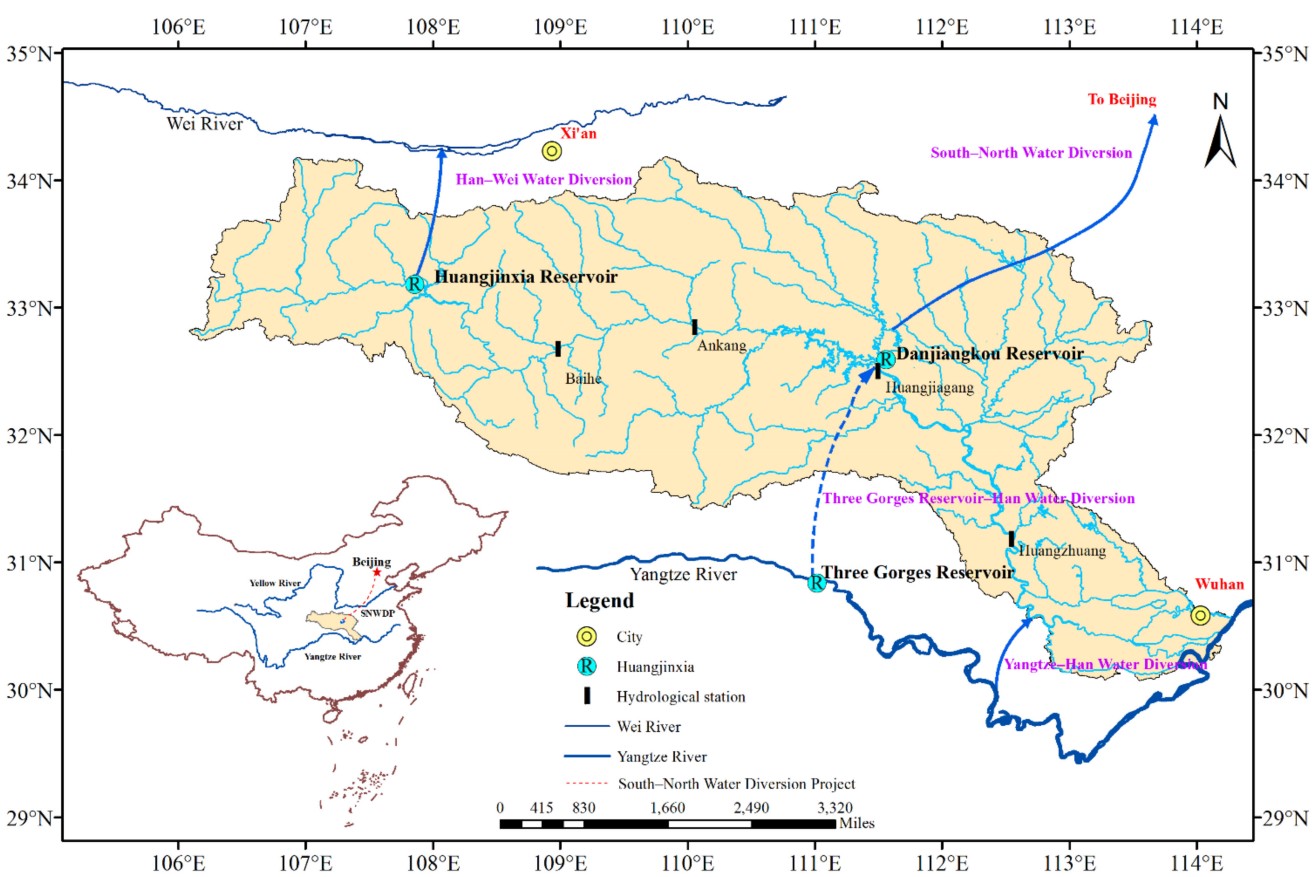

**Figure 1.** Geographic location of the Han River basin and water division projects.

## 2.2. Water Diversion Project

The Han River basin has affluent natural resources with a strong economic base, occupying a crucial strategic position and playing a prominent role in the economic and social expansion pattern of the basin. The Danjiangkou Reservoir acts as the water source of the South-to-North Water Diversion Middle Route Project in China, and it is responsible for the water supply to Beijing and Tianjin metropolitans and Henan and Hebei provinces [30,31]. The Yangtze-to-Han River Water Diversion Project is a complementary project of the South-

to-North Water Diversion Project, which plays parts in recharging the downstream and improving conditions for industrial and agricultural water consumption, river navigation, and ecological water consumption in the lower reaches of the Han River. Besides, the Han-to-Wei River Water Diversion Project under construction targets to address the water shortage problems in the Guanzhong Plain in Shanxi province [32].

The large-scale interbasin water diversion projects under construction and in operation have great impacts on the water resources in this basin, so they will be taken into consideration in this study. Besides, the Middle Line South-to-North Water Diversion Project, Han-to-Wei River Water Diversion Project, and Yangtze-to-Han River Water Diversion Project [33] also matter. Hence, we set that the scales of the above water diversion projects in the planning year 2035s will be 11 billion, 1.5 billion, and 3 billion m$^3$, respectively.

### 2.3. Data Description

Various data (Table 1) are in the running to drive the SWAT model, including digital elevation model, land use/land cover map, soil map, meteorological data, river discharge, projected precipitation and temperature, and socioeconomic and environmental data. We derive the digital elevation model from the United States Geological Survey (USGS), and the land use/land cover map and soil map are taken from the Resource and Environment Science and Data Center (RESDC) and Food and Agriculture Organization (FAO), respectively. The meteorological data from 1961 through 2005 can be accessed from the China Meteorological Administration. The river discharge data are obtained from the Changjiang Water Resources Commission, Ministry of Water Resources of China. To project future hydro-climatologic scenarios, the multimodel bias-corrected precipitation and temperature data under RCP (Representative Concentration Pathway) 4.5 obtained from Shen et al. [34] are forced to drive the SWAT model. The data required for the development and utilization of water resources mainly come from the China Water Resources Bulletin (2010–2016), Water Resources Bulletin (2010–2016), and Environmental Bulletin of six metropolitans or provinces (2010–2016), also listed in Table 1.

**Table 1.** Data descriptions and sources in this study.

| NO. | Data | Source | Relevant Characteristics |
|-----|------|--------|--------------------------|
| 1 | Digital elevation model | USGS | 90 × 90 m spatial resolution |
| 2 | Land use/land cover map | RESDC | 1 × 1 km spatial resolution |
| 3 | Soil map | FAO | 1 × 1 km spatial resolution |
| 4 | Meteorological data | China Meteorological Administration | 1961–2005 |
| 5 | River discharge | Changjiang Water Resources Commission, Ministry of Water Resources of China | 1980–2000 and 2010–2016 |
| 6 | Projected precipitation and temperature | 20 global climate models (GCMs) [34] | RCP 4.5 |
| 7 | Socioeconomic and environmental data | China Water Resources Bulletin, Water Resources Bulletin, and Environmental Bulletin | Six metropolitans or provinces involved in this study |

## 3. Methodology

The comprehensive evaluation system is constituted by three portions (i.e., water resources subsystem, socioeconomic subsystem, and eco-environment subsystem). As the prediction of the index involved in different subsystems was complex and synthesized, both the SWAT model and water resources development and utilization model were used for predicting future scenarios. The SWAT model was used to simulate and project the runoff and water resources condition based on projected precipitation and temperature series. The water resources development and utilization model was used to project the water consumption of socioeconomic development in the planning year, and its inputs consisted of an economic and social development plan and water consumption level. The framework of this paper is shown in Figure 2.

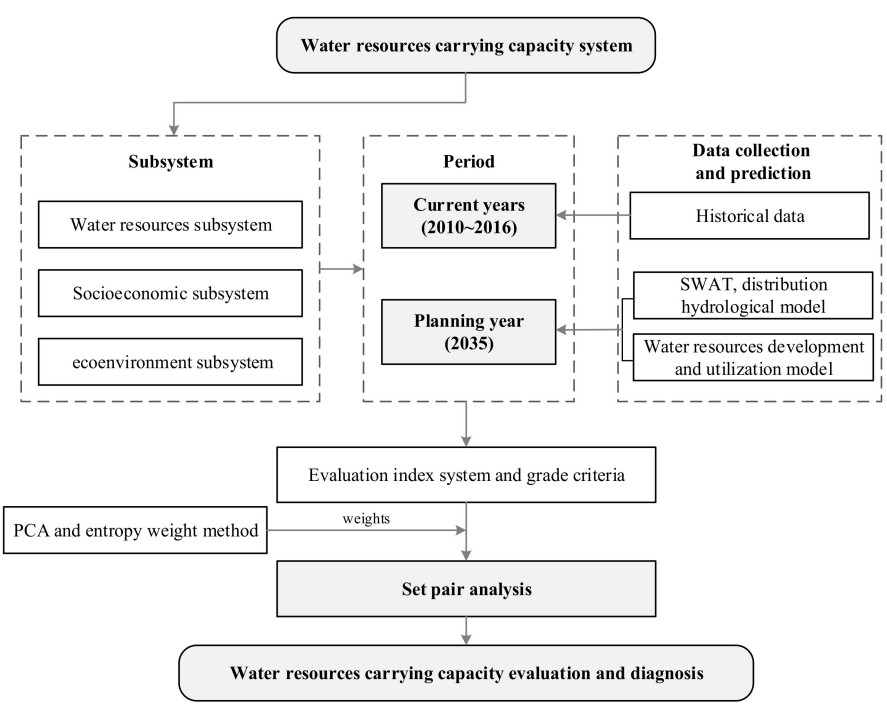

**Figure 2.** Framework of assessing and predicting the water resources carrying capacity (WRCC).

### 3.1. Evaluation Index System

Based on the principles of comprehensiveness, hierarchy, and simplicity, the evaluation index system was built by using the frequency statistical approach and theoretical analysis method. Additionally, internal and external conditions, such as the spatiotemporal distribution of water resources and overall plan of socioeconomic development, were considered. The system consisted of the water resources subsystem, socioeconomic subsystem, and eco-environment subsystem. Twelve indices were finally selected in consonance with the nature and connotation of the problems (Table 2).

**Table 2.** Evaluation index system of water resources carrying capacity in the Han River basin.

| Object Hierarchy | Rule Hierarchy | Index Hierarchy | Index Description |
|---|---|---|---|
| Water resources carrying capacity in the Han River basin | Water resources subsystem | $X_1$: Modulus of water resources production | The amount of water resources |
| | | $X_2$: Modulus of water supply | The level of water supply capacity |
| | | $X_3$: Rate of water resources exploitation and utilization | The level of water resources utilization |
| | | $X_4$: Water resources per capita | The level of water resources per capita |
| | Socioeconomic subsystem | $X_5$: GDP per capita | The level of economic development |
| | | $X_6$: Population density | The population carrying status |
| | | $X_7$: Urbanization rate | The level of urbanization |
| | | $X_8$: Water consumption per $10^4$ yuan GDP | The water consumption |
| | | $X_9$: Daily domestic water consumption per capita | The level of water consumption for population |
| | | $X_{10}$: Water consumption per $10^4$ yuan of industrial added value | The level of industrial development |
| | Eco-environment subsystem | $X_{11}$: Rate of ecological water consumption | The level of ecological environment |
| | | $X_{12}$: Wastewater discharge per area | The pollution status of the water environment |

Principal component analysis [35,36] is a multivariate statistical method that converts multiple unrelated indices into a few independent comprehensive indices. It possesses traits that can simplify complicated problems, render the problem analysis easier and more convenient, get more scientific results, and so forth. Entropy judges the level of valid information contained in the data by reacting to the disorder of the system [37]. The entropy weight method is characterized by a positive relationship between the degree of disorder in the performance of an index and the entropy value, and an inverse relationship with the amount of information it can respond to [38]. Hence, PCA and the entropy weight method were mixed to determine the weight. Figure 3 is a chord diagram that shows the relationship between the selected 12 indices. If these indices belong to one subsystem, they will be converted into one point. The arc length of each index represents its corresponding weight. The specific weight value of each index is listed in Table 3. Once the index system is established, each index needs to be analyzed to establish its reasonable value range and grading standard. The water carrying capacity of the study area was classified into five states, and an individual index was divided into five levels correspondingly (Table 3). The grade classification and reflective meaning are illustrated in Table 4.

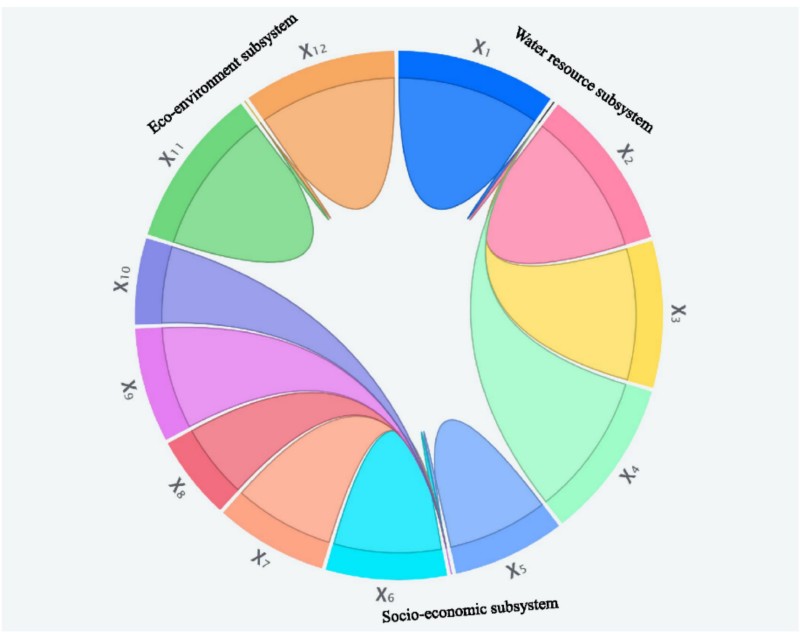

**Figure 3.** Relationship between subsystems and indices with their weights.

**Table 3.** Estimated evaluation index weights and classification standards of the WRCC in the Han River basin.

| Subsystem | Index | Unit | Weighted Value | Carrying Grade | | | | |
|---|---|---|---|---|---|---|---|---|
| | | | | Grade I | Grade II | Grade III | Grade IV | Grade V |
| Water resources subsystem | $X_1$ | $10^4\ m^3/km^2$ | 0.1018 | >50 | 40~50 | 30~40 | 20~30 | <= 20 |
| | $X_2$ | $10^4\ m^3/km^2$ | 0.1032 | <1 | 1~3 | 3~10 | 10~15 | >15 |
| | $X_3$ | % | 0.0934 | <5 | 5~25 | 25~35 | 35~45 | >45 |
| | $X_4$ | $m^3/cap$ | 0.1015 | >3000 | 1700~3000 | 1000~1700 | 500~1000 | <500 |
| Socioeconomic subsystem | $X_5$ | $10^4\ yuan/cap$ | 0.0715 | >5 | 2.5~5 | 1~2.5 | 0.4~1 | <0.4 |
| | $X_6$ | $Population/km^2$ | 0.0768 | <110 | 110~150 | 150~200 | 200~250 | >250 |
| | $X_7$ | % | 0.0715 | <15 | 15~30 | 30~50 | 50~60 | >60 |
| | $X_8$ | $m^3/10^4\ yuan$ | 0.0529 | <90 | 90~110 | 110~250 | 250~600 | >600 |
| | $X_9$ | L/d | 0.0727 | <100 | 100~150 | 150~200 | 200~300 | >300 |
| | $X_{10}$ | $m^3/10^4\ yuan$ | 0.0546 | <8 | 8~10 | 10~15 | 15~20 | >20 |
| Eco-environment subsystem | $X_{11}$ | % | 0.1024 | >4 | 3~4 | 2~3 | 1~2 | <1 |
| | $X_{12}$ | $10^4\ tons/km^2$ | 0.0976 | <1.5 | 1.5~2.0 | 2~2.5 | 2.5~3.0 | >3.0 |

**Table 4.** Grade classification and state description.

| Grade | State | State Description |
|:---:|:---:|:---:|
| I | Best | The situation of the system is optimistic. |
| II | Better | The water resources can furnish better guarantee. |
| III | General | The system remains in a relatively stable state. |
| IV | Worse | There is a certain degree of guarantee with limited potential. |
| V | Poor | The system is severely at risk. |

### 3.2. Set Pair Analysis

A set pair analysis put forward by Zhao [39] was primarily adopted to grapple with uncertainty problems [40]. It is well-known for its ease of implementation, qualitative and quantitative analyses, and uncertainty consideration. Owing to the variability and complexity of the water resources, there is a complicated and uncertain relationship between each evaluation factor and the carrying capacity level [41].

The main idea is to construct the research into two sets with a certain connection, and then systematically analyze the characteristics of the two sets in identity, discrepancy, and contradistinction using the degree of connection for quantitative description. Assuming set $A = (X1, X2, X3, \ldots XN)$ and set $B = (P1, P2, P3, \ldots PN)$ construct a set pair $J = (A, B)$, the expression describing the connection degree can be stated as

$$\mu(A \sim B) = \frac{U}{N} + \frac{V}{N}i + \frac{W}{N}j \tag{1}$$

where $\mu(A - B)$ refers to the connection degree of sets $A$ and $B$; $N$ refers to the total number of elements; $U$ refers to the number of identical elements, in which identity signifies that sets $A$ and $B$ are identical; $V$ refers to the number of discrepant elements, in which discrepancy signifies that there are some subtle differences between sets $A$ and $B$; $W$ refers to the number of contradictory elements, in which contradistinction signifies that remarkable differences exist in sets $A$ and $B$; accordingly, $U/N$, $V/N$, and $W/N$ refer to the identical degree, discrepancy degree, and contrary degree, respectively; $i$ refers to the uncertain coefficients of the discrepancy degree valuing in the range of $-1$ and $1$; and $j$ refers to the coefficient of the contrary degree and generally takes the value of $-1$, playing the role of contrary mark sometimes. Setting $a = U/N$, $b = V/N$, and $c = W/N$, then Equation (1) can be written as

$$\mu(A \sim B) = a + bi + cj \tag{2}$$

where the values of the coefficients satisfy $a + b + c = 1$. Equations (1) and (2) are the connection degrees that are commonly used (i.e., the three-element connection degree). $bi$ in Equation (2) can be expanded to $bi = b_1i_1 + b_2i_2 + \cdots + b_ki_k$, so a multi-element connection degree can be attained.

A set pair $H(A_l, B_k)$ can be formed when the values of an index $x_l$ ($l=1, 2, 3, \ldots, m$; $m$ denotes the number of evaluation indices) in the evaluation are viewed as one set called $A_l$ and the evaluation criteria for the corresponding index are taken as another set $B_k$ ($k=1, 2, 3, \ldots, K$; $K$ denotes the number of evaluation levels). Based on the principle of SPA, the $K$-element connection degree of $H(A_l, B_k)$ can be defined as

$$\begin{aligned} \mu = \mu_{A_l \sim B_k} &= \sum_{l=1}^{m} w_l \mu_l \\ &= \sum_{l=1}^{m} w_l a_l + \sum_{l=1}^{m} w_l b_{l,1} i_1 + \sum_{l=1}^{m} w_l b_{l,2} i_2 + \cdots + \sum_{l=1}^{m} w_l b_{l,K-2} i_{K-2} + \sum_{l=1}^{m} w_l c_l j \end{aligned} \tag{3}$$

where $w_l$ refers to the weight of the $l$th index, which can be assigned due to its contribution to $\mu$.

Let $S_1,S_2,S_3,S_4,S_5$ be the thresholds for each index from Grades I to V, respectively, and the connection degree $\mu_l$ of the sample $x_l$ with its evaluation criteria at Grade I can be expressed as the line in Figure 4. Then, the confidence criterion is utilized to judge the ranks of sample.

$$h_k = (f_1 + f_2 + \cdots + f_k) > \lambda, k = 1, 2, \cdots 5 \tag{4}$$

in which

$$f_1 = \sum_{l=1}^{m} w_l a_l, f_2 = \sum_{l=1}^{m} w_l b_{l,1}, \cdots, f_{K-1} = \sum_{l=1}^{m} w_l b_{l,K-2}, f_K = \sum_{l=1}^{m} w_l c_l \tag{5}$$

where $\lambda$ refers to the confidence level. The greater the value of $\lambda$ is, the more conservative and safer the evaluation result is. $h_k$ refers to the sum of the first $K$-elements in the connection degree, and $f_1$, $f_2$, $f_3$, $f_4$, $f_5$ refer to the identical degree, partial identical discrepancy degree, uncertainty discrepancy degree, partial contrary discrepancy degree, and contrary degree, respectively.

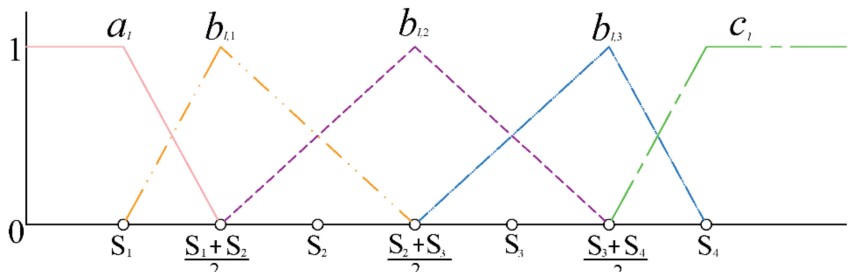

**Figure 4.** The sketch map of the degree of connection in set pair analysis (SPA).

The set pair potential function of the connection degree is its adjoint function, which means the relative deterministic state and development trend of the study object at the macro level. Subtraction set pair potential (SSPP) is used for situation analysis [42]. Based on the identity, discrepancy, and contradistinction of SPA, SSPP can be defined as

$$S_f(u) = (a - c)(1 + b) \tag{6}$$

where the scope of the value of $S_f(u)$ is $[-1.0, 1.0]$. $a$, $b$, and $c$ refer to the same meaning in Equations (1) and (2). $S_f(u)$ is divided into five classes as inverse potential ($S_f(u) \in [-1.0, -0.6)$), partial inverse potential ($S_f(u) \in [0.6, -0.2)$), symmetrical potential ($S_f(u) \in [-0.2, 0.2]$), partial identical potential ($S_f(u) \in (0.2, 0.6]$), and identical potential ($S_f(u) \in (0.6, 1.0]$). The major element degenerating the carrying status is what belongs to the inverse potential or partial inverse [43]. Diagnosed as the vulnerability index, this index serves as the chief object for the regulation of the carrying capacity.

### 3.3. Distributed Hydrological Model

The SWAT model is a geographic information system (GIS)-based distributed hydrological model with a clear physical mechanism [44]. The model can take advantage of GIS to extract a digital elevation model to form flow networks in sub-basins for simulating the hydrological cycle process. The model typically divides watersheds into several sub-basins with different soil types and land use attributes [45]. Water balance is the premise and root of the hydrological cycle simulation in SWAT, the equation of which can be expressed as

$$SW_t = SW_0 + \sum_{i=1}^{t}(R_{day,i} - Q_{surf,i} - E_{a,i} - W_{seep,i} - Q_{gw,i}) \tag{7}$$

where $SW_t$ refers to the final soil water content (mm), $SW_0$ refers to the initial soil water content (mm), $t$ refers to the time with units of days, $R_{day,i}$ refers to the precipitation on the $i$th day (mm), $Q_{surf,i}$ refers to the surface runoff on the $i$th day (mm), $E_{a,i}$ refers to the evapotranspiration on the $i$th day (mm), $W_{seep,i}$ refers to seepage from the soil profile on the $i$th day (mm), and $Q_{gw,i}$ refers to the underground runoff on the $i$th day (mm). Predominantly, the coefficient of the Nash–Sutcliffe efficiency *(NSE)* and the relative error *(RE)* are exploited to evaluate the results of the SWAT simulation.

$$NSE = 1 - \frac{\sum\limits_{t=1}^{n}\left(Q_0^t - Q_s^t\right)^2}{\sum\limits_{t=1}^{n}\left(Q_0^t - \overline{Q_0}\right)^2} \tag{8}$$

$$RE = \frac{\sum\limits_{t=1}^{n}\left(Q_0^t - Q_s^t\right) \times 100\%}{\sum\limits_{t=1}^{n} Q_0^t} \tag{9}$$

where $Q_0^t$ represents the observed discharge at time $t$, $Q_s^t$ represents the simulated discharge at time $t$, $\overline{Q_0}$ represents the mean of observed values, and $n$ represents the number of observed data. On the condition that the results satisfy $NSE > 0.5$ and $RE < 15\%$ during both calibration and validation periods, the SWAT model is applicable for this basin, and the simulation results are acceptable.

In this paper, the observed discharge data from four hydrological stations, including Ankang, Baihe, Danjiangkou, and Huangzhuang, were exploited for calibrating and validating the SWAT model. The Ankang and Baihe hydrological stations are located in the upper reaches of the basin, while the Danjiangkou hydrological station is in the middle reaches, and the Huangzhuang station lies in the lower reaches. The calibration began on 01 January 1980 and lasted until 31 December 1993, and validation was performed during the period 01 January 1994–31 December 2000.

### 3.4. Water Resources Development and Utilization Model

To capture the dynamic properties of the water resources subsystem, socioeconomic subsystem, and eco-environment subsystem, a compiled model was used to predict the values of evaluation indices scientifically in the planning year.

Water resources

Modulus of water resources production refers to annual water resources amount per unit area [46]. The total water resources amount of the study area can be obtained according to the simulation result of the SWAT model. The corresponding formula for computing the entire amount of water resources is as follows:

$$W = R_s + P_r = R + P_r - R_g \tag{10}$$

where $W$ denotes the entire amount of water resources ($m^3$), $R_s$ denotes the surface runoff (i.e., the difference between streamflow and baseflow, $m^3$), $P_r$ denotes the precipitation infiltration quantity ($m^3$), $R$ denotes the streamflow (i.e., the surface water resources amount, $m^3$), and $R_g$ denotes the baseflow ($m^3$).

Water supply and consumption

Water consumption refers to the sum of the water used by all types of off-stream water users, including losses from water transmission [47]. The quota method was applied to assess the water consumption, the explicit steps of which are as follows: Step 1: explore the trends of the main factors affecting water consumption and determine water consumption indices and quotas. Step 2: compute the amount of water consumption in the planning year in terms of the indices and quotas derived from Step 1, such as population and industrial production. In the light of the classification of water users, the water consumption of each

sector can be estimated based on the influence factors and quotas. The gross amount of water consumption refers to the sum of water consumption of each sector. Meanwhile, the estimated amount of water supply is equal to water consumption. The formula for estimating water consumption is as follows:

$$WD = WQ \cdot WA / (1 - LR) \tag{11}$$

where $WD$ refers to the water consumption of one sector, $WQ$ refers to the water quota per unit of one sector, and $WA$ and $LR$ refer to water use per activity level and the water transferring loss ratio of the sector, correspondingly. For example, when we estimate the domestic water consumption, $WD$ refers to the amount of domestic water consumption (m$^3$), $WQ$ refers to the water consumption per person (m$^3$/person), $WA$ refers to the total population (number of persons), and $LR$ refers to the water transferring loss ratio of domesticity. As for the water consumption of an industry, agriculture, and ecological environment, the parameters represent the corresponding water consumption indices and quotas.

Water pollution and water environment

The rate of ecological water consumption is calculated by the outputs from the water consumption in the water resources development and utilization model. The wastewater discharge is calculated based on the water consumption of a domesticity and industry with the corresponding pollution discharging coefficients.

$$X_{11} = WD_{eco} / WD_{total} \tag{12}$$

$$X_{12} = \alpha \cdot WD_{dom} + \beta \cdot WD_{ind} \tag{13}$$

where $WD_{eco}$ refers to the water consumption of an ecology, and $WD_{total}$ refers to the sum of water consumption of all sectors. $WD_{dom}$ and $WD_{ind}$ refer to the water consumption of a domesticity and industry, respectively. $\alpha$ and $\beta$ refer to the pollution discharging coefficient of a domesticity and industry, respectively. Historical data from the Statistical Bulletin of the National Economic and Social Development and Environmental State Bulletin of the cities in the Han River basin are allowed for this study. Besides, the industrial structures, development trends, and environmental protection requirements are considered to predict the value comprehensively.

## 4. Results

### 4.1. Evaluation of the Current Water Resources Carrying Capacity

The WRCC during the period 2010–2016 is shown in Figure 5. The calculation of the connection degree of each index is based on Figure 4. The connection degree of the whole system and three subsystems can be attained by combining with the weight of each index. The WRCC was maintained at Grade IV when $\lambda$ was set to 0.75. Meanwhile, the condition was detrimental to facilitating the expansion of the industry. Further analysis of five individual elements illustrated that the value of $a$ slid from 0.21 to 0.10 and then returned up to 0.19, while the value of $c$ remained unchanged. The value of $b_1$ was on the wane, but $b_2$ and $b_3$ showed a growing trend, which implied that the WRCC condition was getting worse.

Table 5 shows that the SSPP belonged to a symmetrical potential from 2010 to 2016. A higher SSPP implies a better carrying condition. The pressure of the WRCC bottomed out in 2010 with a corresponding SSPP value of 0.067; on the contrary, it reached its peak in 2013 with a corresponding value of −0.069. The WRCC in the current years could support the socioeconomic development in this basin. Although the WRCC in 2010–2016 pertained to a symmetrical potential from the perspective of situation analysis, the result of the SSPP fluctuated to a certain extent.

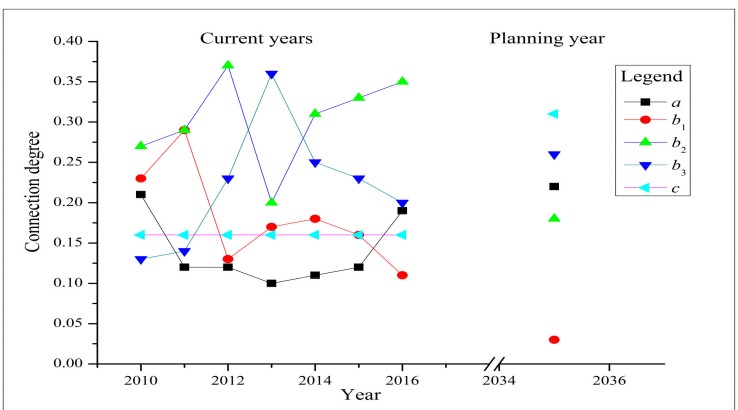

**Figure 5.** The five-element connection degree of the WRCC.

**Table 5.** WRCC connection and subtraction set pair potential.

| | Year | Five-Element Connection Degree | | | | | SSPP | Potential |
|---|---|---|---|---|---|---|---|---|
| | | $a$ | $b_1$ | $b_2$ | $b_3$ | $c$ | | |
| Water resources carrying capacity | 2010 | 0.21 | 0.23 | 0.27 | 0.13 | 0.16 | 0.067 | Symmetrical |
| | 2011 | 0.12 | 0.29 | 0.29 | 0.14 | 0.16 | −0.041 | Symmetrical |
| | 2012 | 0.12 | 0.13 | 0.37 | 0.23 | 0.16 | −0.047 | Symmetrical |
| | 2013 | 0.10 | 0.17 | 0.20 | 0.36 | 0.16 | −0.069 | Symmetrical |
| | 2014 | 0.11 | 0.18 | 0.31 | 0.25 | 0.16 | −0.060 | Symmetrical |
| | 2015 | 0.12 | 0.16 | 0.33 | 0.23 | 0.16 | −0.040 | Symmetrical |
| | 2016 | 0.19 | 0.11 | 0.35 | 0.20 | 0.16 | 0.036 | Symmetrical |
| | mean | 0.14 | 0.18 | 0.30 | 0.22 | 0.16 | −0.022 | Symmetrical |
| | 2035s | 0.22 | 0.03 | 0.18 | 0.26 | 0.31 | −0.100 | Symmetrical |
| Water resources subsystem | 2010 | 0.20 | 0.44 | 0.24 | 0.12 | 0 | 0.250 | Partial identical |
| | 2011 | 0 | 0.47 | 0.39 | 0.14 | 0 | 0 | Symmetrical |
| | 2012 | 0 | 0 | 0.66 | 0.34 | 0 | 0 | Symmetrical |
| | 2013 | 0 | 0 | 0.33 | 0.67 | 0 | 0 | Symmetrical |
| | 2014 | 0 | 0 | 0.64 | 0.36 | 0 | 0 | Symmetrical |
| | 2015 | 0 | 0 | 0.70 | 0.30 | 0 | 0 | Symmetrical |
| | 2016 | 0 | 0.04 | 0.77 | 0.19 | 0 | 0 | Symmetrical |
| | mean | 0.03 | 0.14 | 0.53 | 0.30 | 0 | 0.036 | Symmetrical |
| | 2035s | 0 | 0 | 0.35 | 0.42 | 0.23 | −0.291 | Partial inverse |
| Socioeconomic subsystem | 2010 | 0.09 | 0.14 | 0.45 | 0.19 | 0.14 | −0.059 | Symmetrical |
| | 2011 | 0.07 | 0.24 | 0.35 | 0.21 | 0.14 | −0.090 | Symmetrical |
| | 2012 | 0.05 | 0.33 | 0.26 | 0.23 | 0.14 | −0.106 | Symmetrical |
| | 2013 | 0.01 | 0.43 | 0.17 | 0.25 | 0.14 | −0.162 | Symmetrical |
| | 2014 | 0.03 | 0.45 | 0.13 | 0.26 | 0.14 | −0.140 | Symmetrical |
| | 2015 | 0.07 | 0.40 | 0.12 | 0.28 | 0.14 | −0.088 | Symmetrical |
| | 2016 | 0.22 | 0.22 | 0.11 | 0.30 | 0.14 | 0.103 | Symmetrical |
| | mean | 0.08 | 0.32 | 0.23 | 0.24 | 0.14 | −0.077 | Symmetrical |
| | 2035s | 0.31 | 0.08 | 0.10 | 0.03 | 0.48 | −0.180 | Symmetrical |
| Eco-environment subsystem | 2010 | 0.49 | 0 | 0 | 0 | 0.51 | −0.024 | Symmetrical |
| | 2011 | 0.49 | 0 | 0 | 0 | 0.51 | −0.024 | Symmetrical |
| | 2012 | 0.49 | 0 | 0 | 0 | 0.51 | −0.024 | Symmetrical |
| | 2013 | 0.49 | 0 | 0 | 0 | 0.51 | −0.024 | Symmetrical |
| | 2014 | 0.49 | 0 | 0 | 0 | 0.51 | −0.024 | Symmetrical |
| | 2015 | 0.49 | 0 | 0 | 0 | 0.51 | −0.024 | Symmetrical |
| | 2016 | 0.49 | 0 | 0 | 0 | 0.51 | −0.024 | Symmetrical |
| | mean | 0.49 | 0 | 0 | 0 | 0.51 | −0.024 | Symmetrical |
| | 2035s | 0.49 | 0 | 0 | 0.39 | 0.12 | 0.412 | Partial identical |

Note: SSPP refers to subtraction set pair potential; "mean" refers to the mean values in 2010~2016.

In terms of the water resources subsystem, the carrying capacity stood at the top in 2010 and gradually declined until 2013. The SSPP belonged to a partial identical potential, which was beneficial to the subsystem in 2010; however, it remained 0 in the following years. Although it belonged to a symmetrical potential, it did not achieve a better performance than the historical period. Specific analysis of the five elements in the connection degree pointed out that the value of $a$ decreased from 0.20 to 0. The value of $b_2$ subordinate to Grade III followed a fluctuating ascending trend, reaching its peak at 0.77 in 2016. The value of $b_3$ climbed from 2010 to 2013 and then declined. It demonstrated that the situation of the water resources subsystem was not optimistic, coincident with the fact that the rate of water resources exploitation and utilization remained relatively high, but the water resources per capita stayed relatively poor.

In terms of the socioeconomic and eco-environment subsystems, the SSPP of the socioeconomic subsystem dropped from $-0.059$ to $-0.162$ and then bounced back to 0.103. $X_5$ and $X_8$ were the two main factors influencing the socioeconomic subsystem. The classification of $X_5$ and $X_8$ rose from Grade III to II. However, $X_6$ and $X_7$ continued to rise, and $X_{10}$ dropped steadily but they remained relatively high, giving rise to the fluctuation trend. The result of the eco-environment subsystem remained unchanged during this period, which had to do with the fixed level of each index in this subsystem intimately.

### 4.2. Performance of the SWAT Model

The parameter calibration is used to find out the simulated value performing best in tone with the observed value. In this study, the Sequential Uncertainty Fitting programme algorithm (SUFI-2) was harnessed to calibrate the hydrological model on the basis of the monthly discharge data from four hydrological stations. The SWAT-CUP software developed by U.S. Department of Agriculture-Agricultural Research Service exported 12 selected parameters for calibration, and the optimal values are listed in Table 6.

**Table 6.** Description of runoff calibration parameters.

| No. | Parameter | Description | Fitted Value |
|---|---|---|---|
| 1 | Alpha_Bf | Baseflow recession constant | 0.5 |
| 2 | Ch_K2 | Effective hydraulic conductivity of channel (mm/hr) | 70.06 |
| 3 | Ch_N2 | Manning's "n" value for the main channel | 0.06 |
| 4 | Cn2 | Moisture condition II curve number | −0.27 |
| 5 | Gw_Delay | Delay time for aquifer recharge (days) | 184.35 |
| 6 | Gw_Revap | Revap coefficient | 0.1 |
| 7 | Gwqmn | Threshold water level in the shallow aquifer for baseflow (mm) | 0.46 |
| 8 | Esco | Soil evaporation compensation factor | 1.06 |
| 9 | Smtmp | Threshold temperature for snow melt (°C) | −3.39 |
| 10 | Sol_BD | Bulk density of the layer (mg/m$^3$) | 0.26 |
| 11 | Sol_Awc | Available water capacity | 0.38 |
| 12 | Sol_K | Saturated hydraulic conductivity (mm/hr) | −0.43 |

The simulation results of each hydrological station are shown in Figure 6, and the evaluation criteria are concluded in Table 7.

Figure 6 shows that the calibration resulted in at least satisfactory model performances in simulating river runoff. The SWAT model shows a fine performance, but the deficiency of simulating maximum discharge and peak values was exposed. Nonetheless, it exhibited quite a good performance of average and minimum discharge rates. Taking the Ankang station as an example, the *NSEs* were 0.93 and 0.83 in the calibration and validation periods, respectively. Meanwhile, the *REs* were 2.4% and 8.1% in the calibration and validation periods, respectively. The *NSE* was greater than 0.8, and the *RE* was lower than 15% for all the stations, implying a good performance of the SWAT model. Nevertheless, due to the damage of watershed hydrological cycle processes triggered by tremendous impacts of human activities, the simulation results were not conspicuously well produced for all. Still and all, the average *NSE* and the average absolute value of *RE* equaled 0.90, 2.8% and

0.76, 6.2% for the calibration and validation periods, respectively. Those indices indicated a good SWAT model performance for streamflow simulation in the Han River basin.

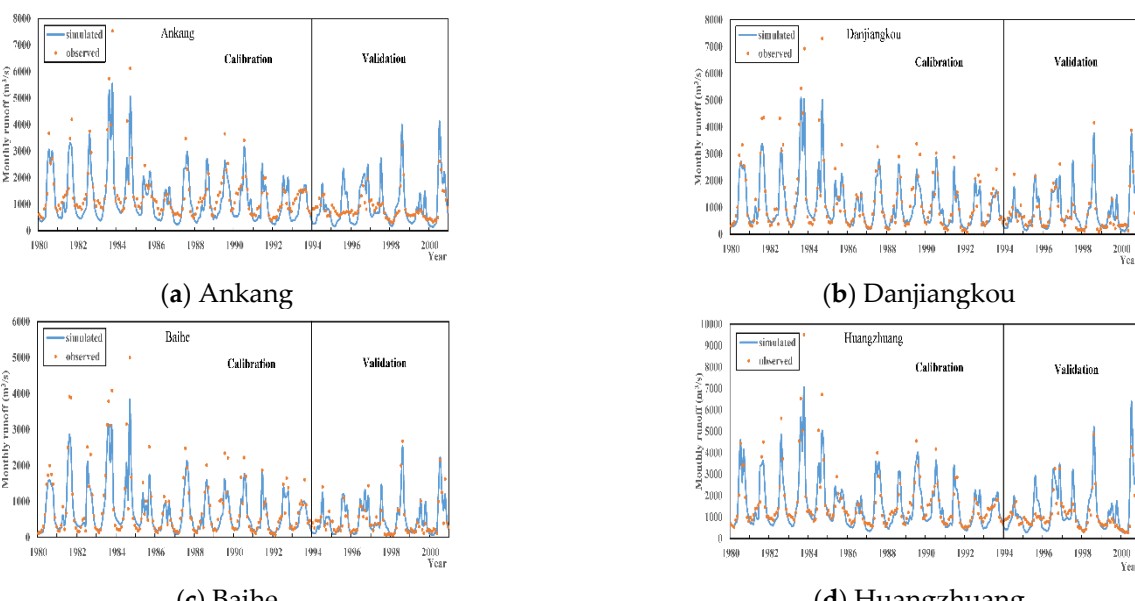

**Figure 6.** The observed and SWAT-simulated monthly streamflow for the calibration period (1980–1993) and for the validation period (1994–2000) in the Han River basin, China.

**Table 7.** The calibration and validation results of the SWAT model.

| No. | Hydrological Station | Calibration Period (1980–1993) | | Validation Period (1994–2000) | |
|---|---|---|---|---|---|
| | | *NSE* | *RE/%* | *NSE* | *RE/%* |
| 1 | Ankang | 0.93 | 2.4 | 0.83 | 8.1 |
| 2 | Baihe | 0.91 | −0.3 | 0.78 | −1.9 |
| 3 | Danjiangkou | 0.92 | 6.9 | 0.75 | 7.5 |
| 4 | Huangzhuang | 0.82 | −1.4 | 0.66 | 7.1 |
| Average absolute value | | 0.90 | 2.8 | 0.76 | 6.2 |

In addition, projected temperature and precipitation have been utilized to analyze climate change impact on hydrological cycle processes under RCP 4.5. As the global climate models' (GCMs) outputs are usually too coarse and biased to represent the spatial heterogeneity of the climatic condition at a basin scale, the daily bias correction method was utilized to correct the GCMs' raw outputs. The bias-corrected temperature and precipitation projections from 20 GCMs were used to force the SWAT model. The ensemble mean streamflow projections from the SWAT is utilized to represent the water resources condition in the future. The water resources during the period 2030–2040 in the study area were acquired from the SWAT model. Compared with the simulated annual mean runoff of 1743 $m^3/s$ in the base period, the runoff in 2035s will reach 1829 $m^3/s$. The results demonstrated that the water resources of the whole study area tended to increase under RCP 4.5.

*4.3. Socioeconomic Projection in Future Periods*

To predict the state of water resources development and utilization in the planning year reasonably, the relevant economic indices and water use level indices need to be determined by comprehensive prediction through different methods.

The statistical analysis of the resident population of the Han River basin illustrated that the average annual growth rate was around 1.82‰ from 2010 to 2016. The growth

rate further expanded to 4.7‰ in 2016. Eventually, the population of each city in 2035s was predicted according to its population planning, contributing to 38 million people in the basin in total. The prediction of the urbanization rate was primarily based on the current urbanization quality and level and strategic development plans. Consequently, the urbanization rate of the Han River basin was predicted to be 66% in 2035s. The Han River basin achieved an average annual GDP growth of 12.44% between 2010 and 2016. With the forward movements of the economic restructuring and the economy's gain in size, economic growth would slow down. Therefore, a target growth rate of approximately 6.5% was set. The added value of the primary industry, secondary industry, and tertiary industry could be acquired in the light of GDP growth rate and the industry structure of each city based on the economic development plans. Factors including cultivated land and water-saving reconstruction were considered. An irrigated area of 1.65 million hectares was predicted for the Han River basin in 2035s.

The projected water resources development and utilization in the planning year is shown in Table 8. The domestic water demands for urban and rural areas are 1.66 billion and 0.48 billion m$^3$, respectively. The water demand for the industry is estimated to be 7.62 billion m$^3$ in total. Meanwhile, the water demands for agriculture and the off-stream eco-environment are 10.52 billion and 0.28 billion m$^3$, respectively. Large-scale water diversion projects will reduce the available water resources enormously. This demonstrates that there will be a water shortage of 2.49 billion m$^3$ in 2035s. Hence, rational utilization of water resources stands out.

**Table 8.** Water resources and water demands in 2010~2016 and in the planning year (billion m$^3$).

| Year | Water Resources | | Water Demands | | | | Water Deficit |
|---|---|---|---|---|---|---|---|
| | Local Water Resources | Available Water Resources | Domesticity | Industry | Agriculture | Ecology | |
| 2010 | 75.95 | 46.61 | 1.26 | 5.59 | 7.65 | 0.05 | 0 |
| 2011 | 67.05 | 37.71 | 1.27 | 5.60 | 8.12 | 0.10 | 0 |
| 2012 | 46.29 | 16.95 | 1.29 | 5.66 | 8.07 | 0.07 | 0 |
| 2013 | 38.92 | 9.58 | 1.30 | 4.58 | 8.83 | 0.07 | 5.19 |
| 2014 | 44.66 | 15.32 | 1.31 | 4.37 | 8.68 | 0.08 | 0 |
| 2015 | 46.72 | 17.38 | 1.37 | 4.50 | 8.82 | 0.09 | 0 |
| 2016 | 49.64 | 20.30 | 1.43 | 4.25 | 8.38 | 0.13 | 0 |
| mean | 52.75 | 23.41 | 1.32 | 4.94 | 8.36 | 0.08 | 0.74 |
| 2035s | 56.90 | 18.06 | 2.14 | 7.62 | 10.52 | 0.28 | 2.49 |

Note: "mean" refers to the mean values in 2010~2016.

*4.4. Evaluation Result in the Planning Year*

The WRCC in the planning year can be estimated using SPA (shown in Table 5). The five elements of the connection degrees will be 0.22, 0.03, 0.18, 0.26, and 0.31. The WRCC will belong to Grade IV if λ is set to 0.75, which means that it will take a toll on urban sprawl or economic growth under overloading circumstances. Compared with the status of the WRCC in the current years, the changes of the five elements indicate that the WRCC has undergone a development towards being overloaded. Although there is a slight increase in *a* value, the value of $b_1$ (partial identical discrepancy degree) will fall sharply, and the value of *c* (contrary degree) will rise drastically. Table 5 shows that the SSPP of the WRCC will be subordinate to the symmetrical potential in 2035s, whose value is the minimum. In comparison with the evaluation results in 2010–2016, the pressure of the WRCC will be further increased in 2035s with a corresponding SSPP of −0.1.

Regarding the water resources subsystem in the planning year, Table 5 shows that it will perform worst during all the evaluation periods. With all of the water diversion projects to be completed and to operate, water diversion inflow and outflow will dramatically change the available water resources. The total water resources will reach 47.4 billion m$^3$ after considering the water diversion, lower than that in 2010–2016. With the population growth and development of the economy, water consumption of various types will increase

to a large extent, translating to an increase in water supply. Particularly, the mean of $X_2$ is equal to 9.46 $10^4$ m$^3$/km$^2$ in the current years, but it will rise by nearly 40%. $X_2$ will climb to 13.22 $10^4$ m$^3$/km$^2$, which is prejudicial to the subsystem. As for the restrictive index, this alteration not only is devastating to the subsystem but also exposes the water resources to more pressure. In respect of the socioeconomic subsystem, the SSPP will decrease compared with that in 2016. It will decrease from $-0.077$ in the current years to –0.18 in 2035s. Specifically, the level of $X_7$ will metamorphose from Grade III to V, and $X_9$ will change from Grade II to III. For the eco-environment subsystem, it will improve a lot compared with the current period. $X_{11}$ will reach Grade IV in 2035s, while it had been at Grade V in 2010–2016; $X_{12}$ will be at Grade I with a further optimized value of 10,500 tons/km$^2$. The indices will work together to improve the carrying capacity.

Figure 7 shows the SSPP of each index in the current years and in the planning year. There were two, one, and three identical potential indices in 2010, 2011–2015, and 2016. Meanwhile, there were two inverse potential indices in 2010–2016. This was consistent with the distribution of the carrying status in these years. Likewise, the numbers of identical and inverse potential indices were consistent with their temporal distributions. Consequently, the number of identical and inverse potential indices could mirror the carrying status basically. In the current years, the SSPP values of $X_{10}$ and $X_{11}$ are less than 0.2, so these two indices should be paid attention to. In 2035s, there will be four inverse potential indices, the number of which will be greater than that of the current years. A higher number of inverse potential indices implies a worse carrying capacity condition. Therefore, the water resources in the Han River basin will face more pressure and challenges in the planning year. In the planning year, $X_5$, $X_8$, and $X_{12}$ will perform best with the same value of 1, indicating that it will have a strong carrying capacity and can well support harmonious development. $X_1$, $X_4$, and $X_9$ will be just average belonging to the symmetrical potential, geared towards the WRCC at the planning stage. The poor performers will be $X_2$ and $X_{11}$ belonging to the partial inverse potential. In contrast, $X_3$, $X_6$, $X_7$, and $X_{10}$ will perform worst and belong to the inverse potential. Therefore, $X_2$, $X_3$, $X_6$, $X_7$, $X_{10}$, and $X_{11}$ will be the vulnerable indices, causing the weak regional water resource carrying capacity at the planning level. Furthermore, $X_7$ and $X_{10}$ will be the most prioritized regulation objects for improving the regional WRCC.

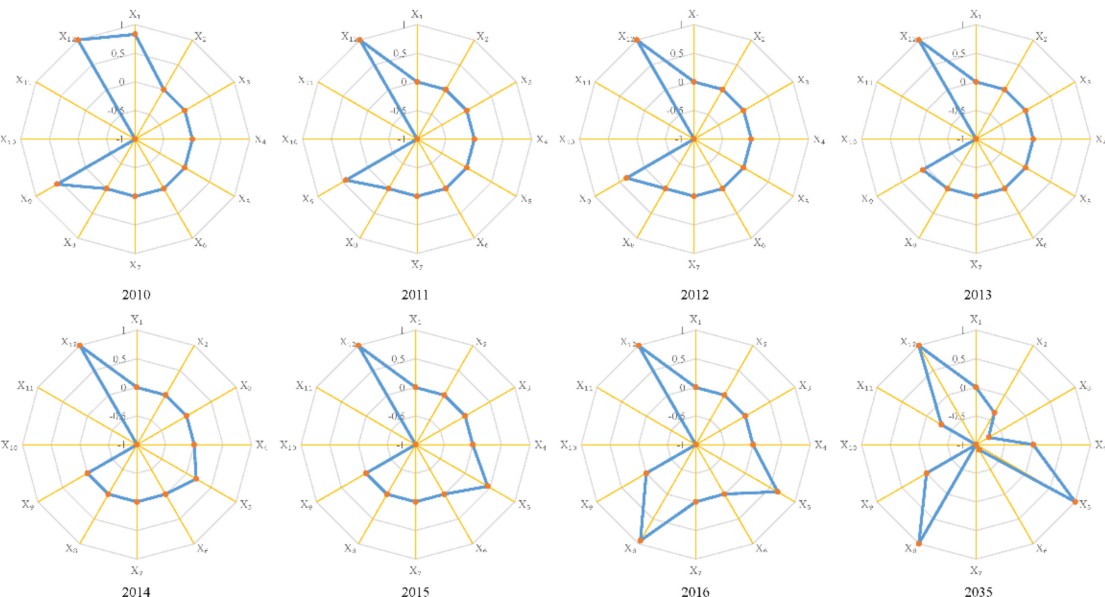

**Figure 7.** Subtraction set pair potential of each index in the current years and in the planning year.

## 5. Discussion

The WRCC is a dramatically essential index measuring coordination between water resources, human beings, and the sustainability of regional economic development. There are numerous collisions between humans and nature over water resources in the Han River basin [33]. To cater to water resources planning, protection, and rational utilization, there is a compelling need to evaluate the WRCC for the national strategic water resources security zone. Previous research studies mainly focused on historical conditions but rarely considered future scenarios under climate changes and social development. Apart from that, the dynamics of the water resources carrying capacity system have not been paid great attention to. This study develops a methodology integrating a hydrological model, projected socioeconomic development model, and evaluation system for assessing and predicting the WRCC of the Han River basin. This methodology not only encompasses three subsystems with features of varying indices but also provides a comprehensive evaluation result considering dynamics. Its application to the Han River basin has proved its superior performance (Tables 5–8 and Figures 5–7). The constructed evaluation index system adheres to the requirements of simplicity, operability, and comprehensiveness and hierarchy. A water resources subsystem, socioeconomic subsystem, and eco-environment subsystem were all included in the evaluation index system. Embedded in the GIS platform in the form of a toolbar, the SWAT model features a convenient operation, user-friendly interface, high degree of structure, ease of use, and so forth. This study divided the total water consumption into four main types (i.e., water consumption for domesticity, industry, agriculture, and ecology). The quota method was employed to accomplish the prediction with a thorough view of the growth in population and economy and water consumption level and condition. The comprehensiveness of SPA was shown in practical experience that it not only considers the identity of two mathematical sets but also mirrors the discrepancy and contradistinction between the sets.

Even though the superior performance of the proposed methodology has been demonstrated in its application to the Han River basin in China, some work still requires to be further explored. The WRCC is a complex system that is affected and interacted by the three major systems of water resources, social economy, and ecological environment. This paper fully considers the influencing factors of the water resources system and the social and economic systems. Due to the obstacle to collecting data, the consideration of the ecological environment system is not remarkably comprehensive, especially in the aspects of water ecology and water environment. With increasing attention being paid to the spells of ecology and the environment by the nation and social public [3], it is bound to be a key issue for future research to comprehensively consider the water environment and water ecology carrying capacity while studying the WRCC.

The SWAT model involves a multitude of parameters that are intricate and arduous to be calibrated. Therefore, the hydrological model needs to be improved, combining specific natural conditions to obtain more desired simulation results. We can call the existing modules and add the required modules according to the conditions, which gains more applicability, prosperity, and reliability to brace for complex hydrological cycle issues [48,49]. Besides, this paper solely puts emphasis on the simulation and projection of discharge in the Han River basin under the RCP 4.5 scenario. Other representative concentration pathways [50] can also be included to simulate discharge under future climate changes to provide more diverse results. The evaluation and diagnostic model based on the connection degree proposed in this paper has good application value and universality in the WRCC. Alongside this, we can try to apply the evaluation and diagnostic model and research ideas established in this paper to land carrying capacity [51], water environmental carrying capacity [52], ecological carrying capacity [53], and other systems.

## 6. Conclusions

A comprehensive methodology is developed to assess the WRCC, and is applied to one of the most essential basins in China. A comprehensive evaluation system coupling

different models is established for the assessment of the WRCC. This system with comprehensiveness and rationality is quite conducive to perceiving water resources condition. The findings of the research are of high significance to advance high-quality development between water resources, social economy, and ecological environment subsystems. This study gives policy-makers a window to digest important information on water resources exploitation and regulation strategies. The main conclusions are as follows:

(1) An evaluation index system including three criterion layers, such as water resources, social economy, and ecological environment, was set up to evaluate the WRCC in the Han River basin, and weight analysis demonstrated that $X_2$ (modulus of water supply) weighed the most with a weight of 0.103, while $X_8$ (water consumption per $10^4$ yuan GDP) accounted for the lightest weight. The SPA manifested that the WRCC in the Han River basin first decreased and then increased. The water resources subsystem operated best with the lowest pressure, while the eco-environment subsystem performed worst in 2010–2016.

(2) For the sake of the prediction of the WRCC in the future, the hydrological model and water resources development and utilization model were coupled to predict the value of each index in the planning year. The SWAT model revealed that the total water resources will reach 47.4 billion $m^3$ after considering the quantity of transferred water in 2035s, which presented lower than that in 2010–2016. The water resources development and utilization model showed that there will be 38 million people with an urbanization rate of 66% in the Han River basin in 2035s, resulting in the water consumption climbing to 20.5 billion $m^3$ in total.

(3) The pressure of the WRCC will further increase, and the system will be confronted with more challenges in 2035s in reference to the current years. The results of the SSPP show that half of the indices will be vulnerability indices weakening the WRCC at the 2035s planning level. The water distribution projects and optimal water resources allocation system should be promoted for construction and implementation to alleviate the grave condition. The methodology integrated the natural water cycle, and water resources management could be utilized to assess and predict the WRCC dynamically and efficiently. Furthermore, the core idea of this paper provided a new tool to evaluate and predict other systems.

**Author Contributions:** Conceptualization and software, L.D. and J.T.; data curation, J.Y. and Q.L.; formal analysis, L.D., J.Y., and J.T.; writing—Original draft preparation, L.D.; writing—Review and editing, J.Y. and S.G. All authors have read and agreed to the published version of the manuscript.

**Funding:** This study was funded by the National Natural Science Foundation of China (Grant Nos. U20A20317, 51879192, and 52009091). It is also partly supported by the Natural Science Foundation of Hubei Province (No. 2020CFB239) and the China Postdoctoral Science Foundation (No. 2020M682478).

**Institutional Review Board Statement:** Not applicable.

**Informed Consent Statement:** Not applicable.

**Data Availability Statement:** The data presented in this study are available on request from the corresponding author. The data are not publicly available due to the raw/processed data required to reproduce these findings cannot be shared at this time as the data also forms part of an ongoing study.

**Acknowledgments:** The authors would like to express their gratitude to anonymous reviewers for their insightful and constructive comments.

**Conflicts of Interest:** The authors declare no conflict of interest.

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
