# Peer review of "Comprehensive Evaluation of Water Resources Carrying Capacity in the Han River Basin"

_water, doi:10.3390/w13030249_

Round 1

Reviewer 1 Report

The authors proposed a framework that integrates hydrological model, water resources development and utilization model model and evaluation system, coupled with the Set Pair Analysis, in quantifying current and projected the water resources carrying capacity (WRCC). The framework was applied to Han River, China. The study showed that under the current condition, the WRCC decreased first but increased later on. However, the water crisis is projected to be more severe in 2035. Overall, the topic of the study is of interest to the readership of the journal. The manuscript is well structured and organized. My comments for the authors include:   Line 26: what does "the risk of WRCC" mean here?   Line 153: should the caption be "Framework of assessing and predicting WRCC"? Does the output from SWAT serve as the input to the Water resources development and utilization model only? or part of its output is only used for the planning year directly?    Table 2 and Table 3: it seems like that x1-4 belongs to the first subsystem (i.e., water resources subsystem); x5-10, the second; and x11-12, the third. Please make it clear.    References 35 and 36 are not the original references for PCA.   More description on Figure 3 is required  (can be put to the appendix if it is necessary to make the main context concise and more focused).
Section 3.4: consider enumerating "Water resources", "Water supply and consumption", and "Water pollution and water environment" using, for instance, 3.4.1, 3.4.2, and 3.4.3, respectively.   Remove "(1)" right before Section 4.1 and Section 4.2   The first sentence in Section 4.1 states that Figure 5 shows WRCC from 2010-2016. In fact the figure also shows 2035 values. Presumably, no WRCC is calculated between 2016 and 2035.If that is the case, please update Figure 5 accordingly.   From Figure 5 and Table 5, the relationships among five element connection degree, Grades I through V, and indices x1 to x12 are not self-explanatory. Please elaborate in detail.    Section 4.3: elaborate how future water pollution and water environment are projected.   Table 5: regarding 2035, only one year in the future is analyzed? or it represents the average of a future period, e.g., 2020-2050?   Table 6: are these parameters consistent across all hydrological simulation units (sub-basins)? if yes, the model is used in a lumped (rather than distributed) way in terms of model parameters.    Section 4.2: "The SWAT projected temperature and precipitation implied that...". Aren't the temperature and precipitation projections coming from GCMs (rather than SWAT)? where GCMs were used? were GCM climate projections downscaled? was the ensemble mean streamflow projections from SWAT utilized?   The SSPP for each index is a critical measure in the study. Please discuss its implications in the discussion section and add relevant findings as an additional main conclusion in the last section.   Description of the results in Section 4.1 and Section 4.4 is not informative. The messages conveyed by those tables/figures are blurred. Consider providing a few concise key messages illustrated by each table/figure.   In the abstract and in Table 8, it states that the total water resources will reach 56.9 billion m3 in 2035; in Section 4.4 and in the Conclusion section, another number 47.4 billion m3 (after water diversion projects) is used. Please describe or illustrate how these numbers are derived.   Last point in the conclusion section: "an overall degradation trend in 2035", be specific about "trend" here as it refers to an overall tendency in a time series. And that the study did not present any relevant results on that.

Reviewer 2 Report

Comprehensive Evaluation of Water Resources 2 Carrying Capacity in the Han River Basin
Global review
In this paper a new methodology integrating hydrological model, climate model, socio-economic development model and dynamic evaluation system is presented to investigate the water resources carrying capacity (WRCC) in the Han River basin, China. This issue is within of scope of “Water”, but a revision is required before publication.
Review comments
Page 2, Line 62: This expression is not appropriate in scientific paper “Meanwhile, other diverse approaches have joined the camp.”
Page 2, Line 71: This expression is not appropriate in scientific paper “… there is still a lack of full digest of extensive problems.”
Page 2, Line 83: See text formatting “… subsystem, socio-economic sub-system, …”
Page 3: Improve Figure 1
Page 4, Line 129: This expression is not appropriate in scientific paper “… are likewise exceedingly front-page.”
Page 4, Line 134: Please, present all data description used: “… soil map and so forth.”
Page 4, Line 138: Please, clarify “Changjiang Water Resources Commission of the Ministry of Water Resources secures the river discharge data.”
Page 4, Table 1: Clarify the meaning of “GCMs”
Page 4, Line 151: This expression is not appropriate in scientific paper “… research process of this paper …”
Page 8: Improve the text “Following the principles of comprehensiveness, hierarchy, simplicity, operability, dynamics and variability, this paper fully considered the characteristics of water resources. The frequency statistical approach and theoretical derivation were used to build the evaluation index system. Additionally, the internal and external conditions including the spatiotemporal distribution of water resources and overall plan of social and economic development were considered. Moreover, the system consisted of water resources subsystem, socio-economic subsystem, and eco-environment subsystem.”
Page 8, Figure 3: Description of Figure 3 is required
Equation (4), (5) and (6): See text formatting. All variables must be described in text.
Page 10: This expression is not appropriate in scientific paper “… to extract digital elevation DEM topographic …”
Equation (7): Clarify the meaning of “t” and “Qgw,i”
Page 11: System or subsystem?: “To capture the dynamic properties of water resources system, socio-economic system and eco-environment system, …”
Page 11: Please, replace the subtopics of section 3.4: i) Water resources; ii) Water supply and consumption; iii) Water pollution and water environment
Page 11: Please, use “… which are as follows: Step 1) explore the trends of the main factors affecting water consumption and determine water consumption indexes and quotas, Step 2) compute the …”
Page 12: And the loss ratio of the sector?: “For example, when we estimate the domestic water consumption, WD refers to the amount of domestic water consumption (m3), WQ refers to the water consumption per person (m3/person), and WA refers to the total number of population (person).”
Page 12: Improve the text “As for industrial, agricultural, and ecological environment water consumption, the parameters show the corresponding meanings estimated by projected future economic development and water management policies.”
Page 12: See heading of section 4.1
Page 14: See heading of section 4.2

Round 2

Reviewer 1 Report

I sincerely thank the authors for taking time to address all the comments. I think the revised manuscript can be aceepted.

Reviewer 2 Report

Minor review comments

Figure 1: The flow direction and hydrological station have the same symbol. Please, clarify.

Line 289: In cited example, include a description for LR.
